# Sensitivity enhancement of homonuclear multidimensional NMR correlations for labile sites in proteins, polysaccharides, and nucleic acids

Mihajlo Novakovic[1], Ēriks Kupče[2], Andreas Oxenfarth[3], Marcos D. Battistel[4], Darón I. Freedberg [4], Harald Schwalbe[3] & Lucio Frydman [1✉]

Multidimensional TOCSY and NOESY are central experiments in chemical and biophysical NMR. Limited efficiencies are an intrinsic downside of these methods, particularly when targeting labile sites. This study demonstrates that the decoherence imparted on these protons through solvent exchanges can, when suitably manipulated, lead to dramatic sensitivity gains per unit time in the acquisition of these experiments. To achieve this, a priori selected frequencies are encoded according to Hadamard recipes, while concurrently subject to looped selective inversion or selective saturation procedures. Suitable processing then leads to protein, oligosaccharide and nucleic acid cross-peak enhancements of ≈200–1000% per scan, in measurements that are ≈10-fold faster than conventional counterparts. The extent of these gains will depend on the solvent exchange and relaxation rates of the targeted sites; these gains also benefit considerably from the spectral resolution provided by ultrahigh fields, as corroborated by NMR experiments at 600 MHz and 1 GHz. The mechanisms underlying these experiments' enhanced efficiencies are analyzed on the basis of three-way polarization transfer interplays between the water, labile and non-labile protons, and the experimental results are rationalized using both analytical and numerical derivations. Limitations as well as further extensions of the proposed methods, are also discussed.

[1] Department of Chemical and Biological Physics, Weizmann Institute of Science, 7610001 Rehovot, Israel. [2] Bruker UK Ltd., Banner Lane, Coventry, UK. [3] Institute for Organic Chemistry and Chemical Biology, Center for Biomolecular Magnetic Resonance, Johann Wolfgang Goethe-University, D-60438 Frankfurt/Main, Germany. [4] Laboratory of Bacterial Polysaccharides, Center for Biologics Evaluation and Research, Food and Drug Administration, 10903 New Hampshire Ave, Silver Spring, MD 20993, USA. ✉email: lucio.frydman@weizmann.ac.il

Two-dimensional (2D) homonuclear NMR correlations[1,2] are an integral part of the tools used to elucidate the structure and dynamics of organic, and biological molecules[3,4]. These correlations can be mediated by chemical exchange or nuclear overhauser effects (NOEs)[5,6], and are probed by monitoring how polarization from one spin reservoir travels to another via dipolar interactions or chemical kinetics[7–12]. Magnetization transfer (MT) within a $J$-coupled spin network as achieved by TOtal Corelation SpectroscopY (TOCSY)[13,14], leads to complementary information based on bond connectivities. Despite being routinely performed these 2D NMR experiments, and particularly NOESY, suffer from relatively low efficiencies, leading to cross-peaks with low signal-to-noise ratio (SNR), and to a need for extensive signal averaging times to improve this SNR. Detection of NOESY and TOCSY cross-peaks becomes even more difficult when involving labile protons ([1]Hs), as information is then "lost" through chemical exchange with the solvent. Hydroxyl [1]Hs in saccharides, amino [1]Hs in proteins and nucleic acids, amide [1]Hs in disordered proteins, and imino [1]Hs in RNA/DNA, are prototypical examples of such challenging systems: when placed in their natural aqueous environment all of these will undergo a rapid exchange with the solvent, which dramatically reduces the efficiency of their intramolecular polarization transfers.

We have recently introduced Looped PROjective SpectroscopY (L-PROSY)[15], an approach that alleviates these problems by regarding these exchange processes as "resets" within the framework of Anti-Zeno Effects[16–18]. Instead of applying a single mixing period for homonuclear transfers that will then reach kinetically-compromised amplitudes, L-PROSY "freezes" these transfers after they begin to act with their (fastest) initial rate, resets the labile [1]H states to their initial conditions by exploiting their exchange with an unperturbed solvent polarization reservoir, and repeats this process multiple times[19,20]. The ensuing "L-PROSY encoding" acts then as a sort of conveyor belt, causing the NOE/TOCSY cross-peaks to grow with the much more favorable rates characterizing their initial buildups, and lasting for as long as either thermodynamic considerations or the recipient's $T_1$ will accommodate them, before performing the latter's signal detection. By selectively addressing only the targeted [1]Hs and avoiding water perturbation L-PROSY exploits some elements of SOFAST NMR[21,22]; at the same time, by its repeated action, it is also reminiscent of CEST-based polarization transfer[23–26]. Despite their sensitivity gains, L-PROSY experiments are still long, requiring traditional $t_1$ evolution periods to build-up multi-dimensional information. L-PROSY acquisitions can also lead to artifacts arising from an incomplete replenishment of the targeted sites' polarization by the solvent, appearing as harmonics of genuine evolution frequencies and/or as anti-diagonal peaks. The present study demonstrates a new approach capable of alleviating both drawbacks while achieving even more complete MTs, which relies on Hadamard-encoded[27–29] selective polarization transfers from the targeted labile [1]Hs. It is shown that, whether involving multiple selective inversions or a continuous saturation procedure, this provides the highest per-scan enhancements we have seen on either conventionally- or L-PROSY-encoded NOESY and TOCSY experiments involving labile or fast-relaxing [1]Hs. When combined with the compressed-sensing advantages and multiplexing provided by Hadamard encoding, gains of *ca.* two orders of magnitude in SNR/unit_time were observed over conventional counterparts. These gains were noted for a variety of biomolecular systems including sugars, nucleic acids, and proteins, illustrating their generality. The physical principles underlying these gains are described on the basis of a simple multi-site exchange model, leading to analytical descriptions that are generalized by numerical calculations, and which reproduce well the experimental observations.

## Results

### Principles of Hadamard-encoded 2D homonuclear correlations on labile sites

Frequency-domain Hadamard spectroscopy has been proposed as a way to replace the conventional $t_1$ evolution increment of 2D time-domain NMR, by employing a "comb" of frequency-selective (polychromatic) radiofrequency (RF) pulses, which directly address peaks in the $F_1$ frequency domain[27–29]. If the frequencies of these peaks are known and well separated, 180° phase shift manipulations based on Hadamard encoding principles, followed by suitable addition/subtraction manipulations (the Hadamard transform), have been demonstrated as means for speeding up the acquisition of 2D NMR data. The present study brings this compressed-sensing scheme to bear within the framework of MT experiments targeting labile [1]Hs in biomolecules; thanks to continuous repolarizations with a replenishing solvent pool, this is shown to substantially improve the SNR in basic NOESY and TOCSY experiments involving such sites. To implement the resulting Hadamard-encoded MT (HMT) schemes, the original NOESY/TOCSY indirect-domain and mixing-time manipulations, were here replaced by either selective polychromatic saturations or looped polychromatic inversion pulses, addressing solely the fast-exchanging, labile [1]Hs. This leads to the experimental scheme depicted in Fig. 1a, where what we denote as the "HMT block" encodes via an "on" or "off" irradiation, the labile [1]Hs according to a Hadamard scheme. This is done while neither the solvent nor the peaks that will eventually receive polarization from the labile sites, are perturbed. The fact that the large water spin reservoir is not perturbed provides a constant repolarization of the labile protons during the encoding process[30,31], effectively shortening these spins' $T_1$s. Concurrently, the fact that the recipients' spins are untouched, prolongs the efficiency of an MT operating through cross-relaxation, $J$-coupling[32] or chemical exchange, up to times lasting on the order of the latter spins' $T_1$. The resulting asymmetry leads to potentially significant SNR/unit_time gains, as illustrated in Fig. 1b with overlaid conventional and HMT-encoded NOESY and TOCSY spectra addressing the hydroxyl sites of myo-inositol, a prototypical saccharide. It follows that whereas in conventional experiments the exchange that labile [1]Hs undergo with the solvent prevent efficient intramolecular spin-coupling/relaxation transfers, in MT the exchange does the opposite—enhancing correlations and magnifying the labile sites' cross-peaks.

The mechanism underlying the resulting enhancements is reminiscent to that in in vivo CEST NMR[24,33–35], and HMT enhancement factors $\varepsilon$ vs NOESY/TOCSY can be computed using a similar theoretical framework. For concreteness, we consider here a NOESY spin dynamics as depicted by the Bloch-McConnell-Solomon equations, incorporating a continuous saturating field targeting a labile [1]H; these are then used to evaluate the magnetization build-up arising from labile→non-labile [1]H cross-relaxation-driven transfers in MT and conventional NOESY scenarios[24,33,36,37]. These calculations are detailed in the Supplementary Information; as shown in Supplementary Equations (1–5), the steady-state solutions describing the MT from the labile [1]H's equilibrium magnetization $M_{z_l}^0$ into the non-labile [1]H state $m_{z_{nl}}^{MT_{ss}}$ can, for a cross-relaxation rate $\sigma$ and saturating field $\nu_1$, be approximated as

$$m_{z_{nl}}^{MT_{ss}} = \frac{\nu_1^2 \sigma}{pq + \nu_1^2(R_{1_{nl}} + \sigma)} M_{z_l}^0 \qquad (1)$$

where $l$ and $nl$ subscripts depict parameters for the labile and non-labile [1]Hs, $p = R_{2_l} + k_{ex}^l - \frac{k_{ex}^l * k_{ex}^w}{R_{2_w} + k_{ex}^w}$ and $q = \left(R_{1_l} + k_{ex}^l + \sigma - \frac{k_{ex}^l * k_{ex}^w}{R_{1_w} + k_{ex}^w}\right)(R_{1_{nl}} + \sigma) - \sigma^2$ are quantities depending

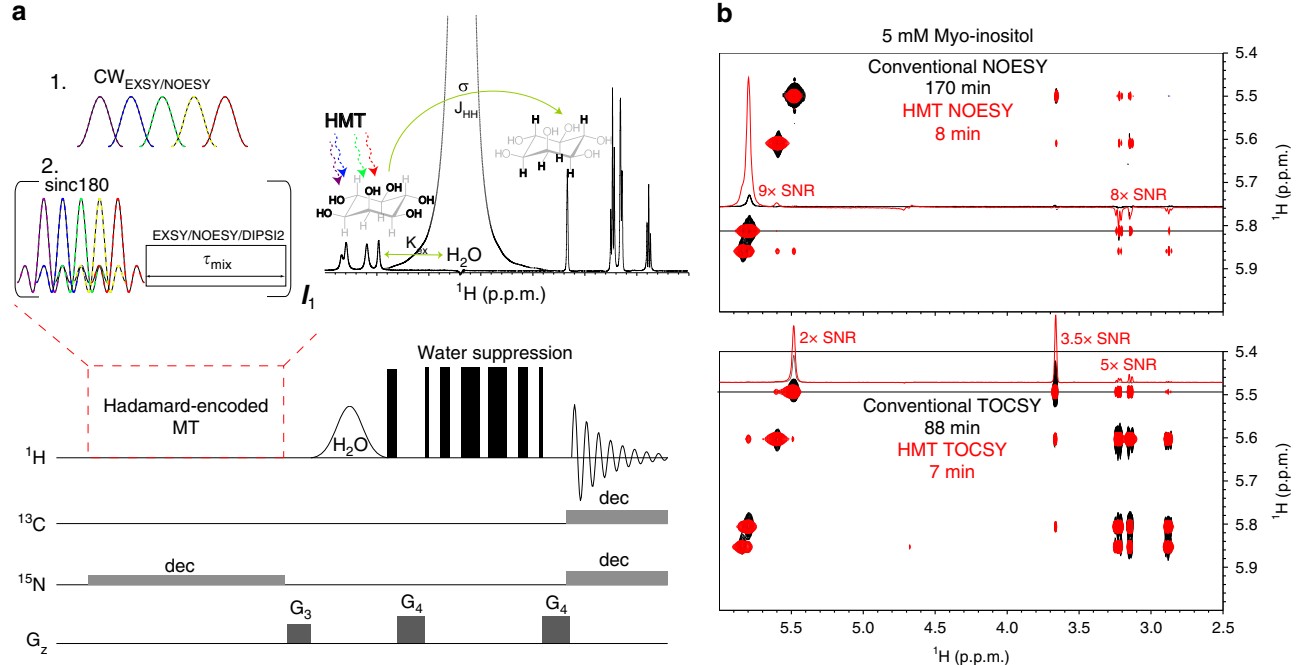

**Fig. 1 Hadamard-encoded magnetization transfer sequences. a** HMT pulse sequence illustrating the two types of perturbation procedures utilized here for improving the efficiency of homonuclear 2D correlations involving labile sites. These were Hadamard-encoded selective polychromatic saturations, or looped Hadamard-encoded polychromatic inversions (assumed here imparted by sinc pulses) followed by a delay for NOESY/EXSY transfer or by DIPSI2[13] to achieve TOCSY's isotropic mixing (selective saturations can be used only for the NOESY/EXSY transfers; repeated inversions work for all). During these long MT processes, illustrated here for myo-inositol, a three-way polarization transfer is effectively established, where water constantly repolarizes the targeted labile ¹Hs, enabling a prolonged magnetization transfer to the non-labile counterparts. "dec" refers to GARP4[68] or adiabatic[69] decoupling used during the encoding and the acquisition for labeled samples; water suppression was achieved using excitation sculpting[70] or WATERGATE 3919[71,62,63] schemes. **b** Comparisons between conventional NOESY (80 ms mixing) and TOCSY (48 ms DIPSI2 mixing) spectra acquired on myo-inositol, and Hadamard MT counterparts (NOESY obtained using 800 ms CW mixing; TOCSY with 12 loops of 24 ms DIPSI2 mixing). Note the different acquisition times required (≈10–20× faster in the HMT scheme), and the substantial (2–9×) enhancements of the cross- and diagonal peaks. Spectra were acquired at 600 MHz on a Bruker Avance III spectrometer equipped with a Prodigy probe.

on the exchange rates $k_{ex}^l$ and $k_{ex}^w$ between the labile site and the water reservoir (scaled by their population ratio), as well as on the longitudinal and transverse relaxation rates $R_{1(2)} = 1/T_{1(2)}$ of the labile, non-labile and water ¹Hs. Similar differential Bloch-McConnell-Solomon equations have been solved analytically for the conventional NOESY experiment[8]; these solutions lead to symmetric $M_{z_l} \to M_{z_{nl}}$, $M_{z_{nl}} \to M_{z_l}$ cross-peaks, reaching maximal values given by:

$$m_{z_{nl}}^{NOESY\,max} = m_{z_l}^{NOESY\,max} = \frac{\sigma}{\lambda_+ - \lambda_-}\left(e^{-\lambda_-\frac{\ln\frac{\lambda_+}{\lambda_-}}{\lambda_+ - \lambda_-}} - e^{-\lambda_+\frac{\ln\frac{\lambda_+}{\lambda_-}}{\lambda_+ - \lambda_-}}\right)M_{z_l}^0$$

(2)

where $\lambda_\pm = \frac{1}{2}\left\{(R_{1_l} + R_{1_{nl}} + 2\sigma + k_{ex}^l) \pm \sqrt{(R_{1_l} + k_{ex}^l - R_{1_{nl}})^2 + 4\sigma^2}\right\}$, and other symbols are as introduced above (see Supplementary Equations (6–10), for a derivation of these maxima conditions). For the labile ¹H scenario of interest here we can assume that $k_{ex}^l \gg R_{1l}, R_{1nl}, \sigma$; this greatly simplifies Eqs. (1) and (2), leading to MT vs conventional NOESY enhancement factors

$$\varepsilon = \frac{m_{z_{nl}}^{MT_{ss}}}{m_{z_{nl}}^{NOESY\,max}} \approx \frac{\nu_1^2 k_{ex}^l T_{1_{nl}}}{(k_{ex}^{l2} + \nu_1^2)}$$

(3)

Notice that according to this model, the enhancement of the labile ¹H's signature will, for an efficient $\nu_1 > k_{ex}^l$ saturation scenario, be given by $k_{ex}^l T_{1_{nl}}$. This represents the number of exchanges that labile ¹Hs can imprint on their cross-relaxing non-labile neighbors given the latter's "memory time", and is

reminiscent of similar behaviors derived for CEST[24,35]. More complete analyses of the predictions deriving from this Bloch-McConnell-Solomon model are presented in the Supplementary Information, including numerical solutions of the labile→non-labile transfer upon selectively perturbing the labile spins' for a variety of solvent exchange rates, couplings and cross-relaxation rates. Supplementary Figures 1–4 confirm that substantial sensitivity improvements may arise from this approach, and examine their dependence on irradiation strength, solvent exchange rate and molecular correlation time. Additional insight is provided by Supplementary Fig. 5, which compares the conceptually similar but practically very different L-PROSY experiment to the current HMT proposal, highlighting the advantages of the latter experiment.

**HMT and the effects of increased magnetic field strengths.** Like CEST[34,38,39], HMT-based methods that target labile ¹Hs could benefit significantly from operating at the highest possible magnetic fields. Under these conditions (i) more rapid exchange rates $k_{ex}^l$ can be accommodated, leading to higher enhancements $\varepsilon$ without resolution penalties (in ppm); (ii) resolution between sites improves, leading to more facile conditions for implementing the 1D Hadamard encoding while relying on more intense $\nu_1$ saturating fields and/or shorter inversion pulses; (iii) $T_{1_{nl}}$ relaxation times tend to get longer, facilitating the extent of the intramolecular transfers[40–42]; and (iv) it becomes generally easier to study the exchanging ¹Hs closer to physiologically-relevant temperatures. These advantages reinforce one another when

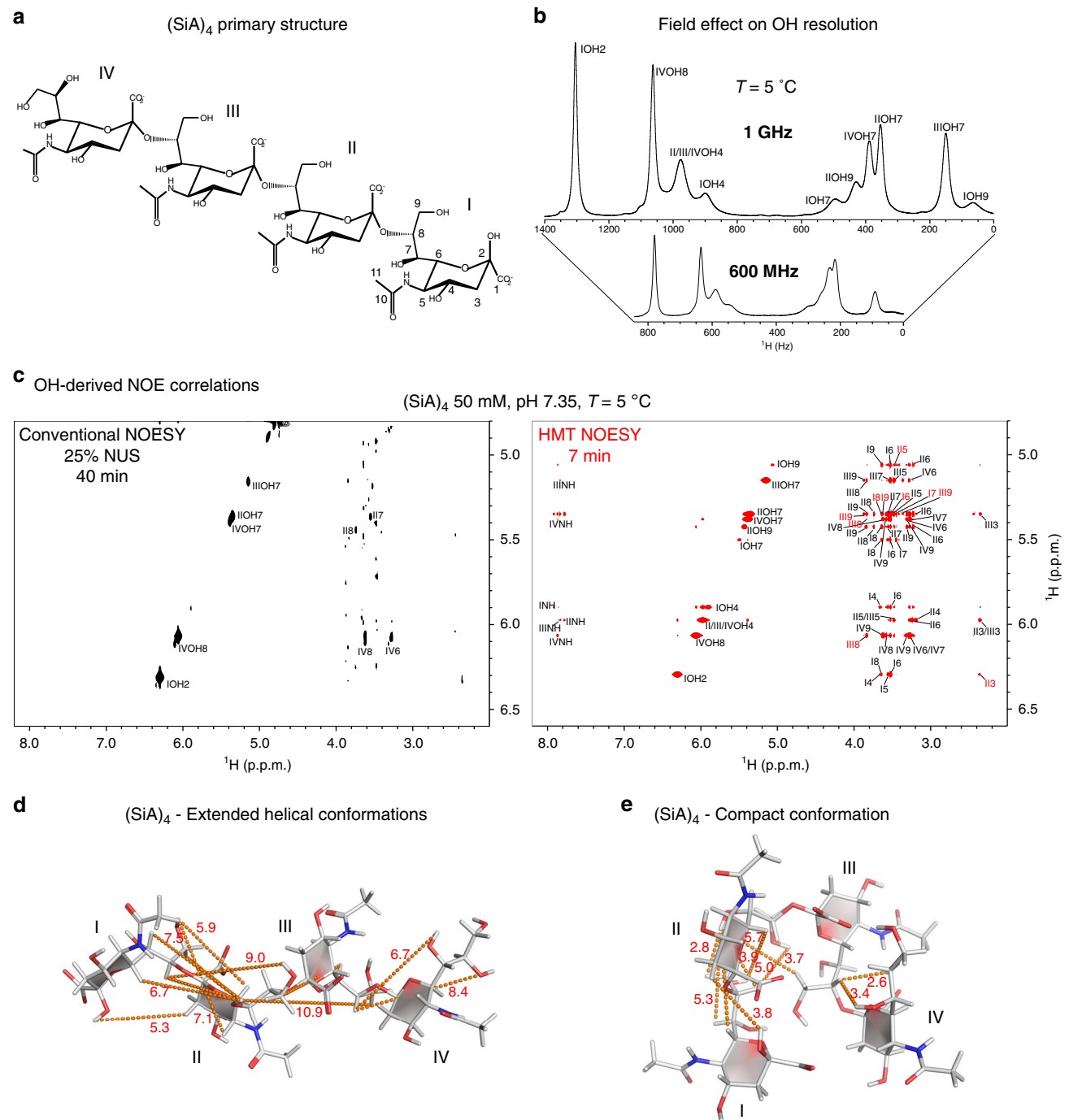

**Fig. 2 HMT applied to the hydroxyl groups of an α2,8-linked sialic acid tetramer. a** Primary structure of (SiA)$_4$, depicting the rings' numbering and the multiple hydroxyl and amide labile $^1$Hs. **b** Magnetic field effect on the appearance of (SiA)$_4$'s hydroxyl $^1$H spectrum. Note how the chemical exchange broadens certain hydroxyl resonances at 600 MHz, whereas at 1 GHz most of the peaks are well-resolved, thanks to field-induced separation of the exchanging sites. **c** Comparison between a conventional 2D NOESY acquired using 25% non-uniform sampling for a minimal acquisition time with faithful spectral reconstruction, and Hadamard MT NOESY collected with a 600 ms saturation period. Although conventional NOESY only reveals some of the closest neighbors, richer information (coupled to much shorter acquisitions) is provided by the HMT. These 1 GHz spectra were acquired using a Bruker Avance Neo console equipped with a TCI cryoprobe; Table 1 summarizes some of the enhancements observed for this compound. **d** Conflicts arising between the cross-relaxation peaks observed in the HMT NOESY spectrum (highlighted in **c** with red fonts and indicated here by dashed lines) and the extended structure proposed for this homopolymer at −10 °C based on non-labile $^1$H NOEs[47]. Notice the long distances (in Å) indicated by the dashed lines for some of the new NOESY-derived experimental connectivities. **e** Revised, compact structure, compatible with the labile $^1$H's NOEs, showing substantial distance shortenings.

targeting rapidly exchanging groups like the OHs of saccharides, something demonstrated in Fig. 2 for the α2,8-linked sialic acid tetramer (SiA)$_4$. Higher fields improve the 1D spectrum for these sites leading, at 5 °C and 1 GHz, to a nearly full resolution of all

hydroxyl $^1$Hs in the spectrum (Fig. 2b). Figure 2c compares conventional and HMT NOESY spectra acquired on (SiA)$_4$ under such conditions; Supplementary Fig. 6a provides similar data but compared at a lower (600 MHz) field. Correlations involving the

**Table 1 SNR enhancements ε extracted upon comparing the various 2D HMT and conventional NOESY and TOCSY spectra collected in this study, for the acquisition times indicated in the corresponding Figures.**

| Biomolecule (labile group) | SNR enhancements ε calculated at the specific chemical shifts (ppm, top rows) | | | | | | $\varepsilon_{average}$$ | $\frac{\varepsilon_{average}}{unit\ time}$ |
|---|---|---|---|---|---|---|---|---|
| Sugars (OH) | Myo-In | 5.5 | 5.6 | 5.8 | 5.9 | – | – | ~8 | ~170 |
| | | 3–10 | 3–10 | 4–10 | >10# | | | | |
| | (SiA)$_4$ NOESY | 5.1 | 5.3 | 5.5 | 5.9 | 6.1 | 6.3 | ~9 | ~50 |
| | | >10# | 2–5 | >10# | >10# | 6 | 4 | | |
| | (SiA)$_4$ TOCSY | 5.1 | 5.3 | 5.4 | 5.5 | 6.0 | 6.1 | ~6 | ~192 |
| | | >10# | 2.3–10 | 1.1–6 | 2.4–10 | 8 | 5–12 | | |
| Protein (NH) | LA5 | 6.7 | 7.3 | 8.0 | 8.3 | 8.8 | 8.9 | ~1.8 | ~10 |
| | | 1–1.7 | 1–2 | 1.7–2.8 | 1.8–2.2 | 1.2–2.8 | 1.7 | | |
| Protein (OH) | Ubiq | 5.8 | 7.4 | 7.5 | 7.6 | 7.8 | 7.9 | ~3 | ~60 |
| | | 1.6–5.1 | 1.2–2 | 1.7–4 | 2–3.3 | 5 | 4.5 | | |
| RNA (NH) | 14mer 1 GHz | 10.9 | 11.7 | 12.7 | 12.9 | 13.2 | 13.8 | ~7 | ~224 |
| | | >10# | >10# | 3–7 | 5–10 | 2.5–4.5 | 4–7 | | |
| | 14mer 600 MHz | 10.9 | 10.9 | 12.7 | 12.9 | 13.2 | 13.8 | ~4 | ~85 |
| | | >5 | >5 | 3–6 | 2.5–5 | 2–3.5 | 2.5–4 | | |

Summarized as well are the average SNR enhancements/unit_time for the different compounds.
#Cross-peak not detected altogether in conventional experiments; a minimum 10-fold enhancement is assumed.
$$\varepsilon_{average}$: average enhancements $(\varepsilon_1+\varepsilon_2+\varepsilon_3+\cdots+\varepsilon_N)/N$ calculated for N peaks in spectrum.

hydroxyl protons are hard to discern in the conventional experiments: even at 1 GHz, the fast chemical exchange reduces NOESY correlations between hydroxyl and aliphatic $^1$Hs to neighbors that are ≤2 Å away, like IVOH8-IV8 (at 1.7 Å) and IVOH8-IV6 (at 1.9 Å). By contrast, HMT NOESY at 1 GHz (and despite the challenges of selective saturation at lower fields, even at 600 MHz) provides a wealth of cross-peaks, in a fraction of the time needed by its conventional counterpart. The fact that in the conventional TOCSY and NOESY spectra many of the cross-peaks are invisible even after protracted acquisitions complicates the quantification of the experimental enhancements; still, Table 1 provides lower bounds for these values. The newly observed correlations involve hydroxyls with a wide rate of solvent exchange rates –ranging from ~10 s$^{-1}$ for IVOH8, to ~40 s$^{-1}$ for IIOH7 and even to ~100 s$^{-1}$ for IIOH9[43]. The reason for this efficiency is as explained earlier: faster exchanges hurt cross-relaxation, but at the same time, supply fresh polarization for the repeated transfer of magnetization (see for instance Supplementary Figs. 1–3). As for the lineshapes displayed by the 2D HMT plots: these are somewhat artificial, as in the Hadamard processing cross-peak traces were simply placed at the frequency positions used for implementing the selective RF irradiation/encoding. In principle, however, the information in the 2D HMT correlations has a well-defined point-spread function, given by 1D absorptive lineshapes along the direct domain, and by T$_1$-, T$_2$-, and $\nu_1$-dependent combinations along the indirect-domain axis. Once again, an analogy to what happens in CEST allows one to compute how these parameters influence the effective F1 spectral peak width[24,35]: this will be given by $\nu_{1/2} \approx \sqrt{\nu_1^2 \frac{p}{q} + p^2}$, where the $p$ and $q$ symbols are as introduced in relation to Eq. (1). Notice that although line-broadening effects are introduced by the saturating RF into these spectral lineshapes, these are less critical than what they may appear at first sight— primarily owing to the labile nature of the $^1$Hs whose 2D correlations these new experiments target. This lability leads to exchange-broadened 1D linewidths, whose spectral resolution is not compromised much further by the RF used in their HMT encoding. In fact, one of the features of HMT is its ability to tailor the saturating field or the inversion bandwidth used in its implementation to each spectral peak individually, enabling an optimal compromise between resolution and the extraction of the information being sought.

Based on a previous (SiA)$_4$ study performed at lower fields under super-cooled conditions[44], the new data arising from HMT NOESY and TOCSY experiments can be used for both assignments and for structural refinements. The latter data are illustrated in Supplementary Fig. 6b, showing a number of sizable intra-residue J-driven correlations (Table 1); in fact, these TOCSY enhancements are often larger than their NOESY counterparts, reflecting perhaps the slightly higher temperatures at which they were measured. The HMT NOESY experiments also reveal correlations that had so far remained unseen, including inter-residue cross-peaks between IOH9-II5/II6, IIOH7-I6/I7/I8/I9, IIOH9-I8, IVOH7-III8, IVOH8-III8. When considering the structure proposed for the glycan under super-cooled conditions[44] a number of conflicts arise, as, according to it, some of the cross-peaks observed by HMT NOESY would then correspond to internuclear distances between $^1$Hs positioned ≈7–9 Å apart (yellow dashes in Fig. 2d). As no strong additional cross-peaks that could justify such long-distance correlations on the basis of a relaying mechanism are visible, this called for a refinement of the glycan's structure. Further investigations to be discussed in further detail separately evidence that, with temperature, (SiA)$_4$ may undergo a rearrangement; this rearranges the oligosaccharide into the average structure shown in Fig. 2e, making it compatible with the observed HMT NOESY cross-peaks.

**Amide, amino, and imino proton correlations in proteins and nucleic acids.** HMT at ultrahigh magnetic field turns out to be especially informative when implemented on the imino protons of nucleic acids. At 1 GHz these imino resonances, which can be broadened by chemical exchange with the solvent at lower fields, are sharper and better resolved. Figure 3 shows the superiority of the HMT experiment over conventional NOESY for detecting cross-peaks involving imino resonances, utilizing a 14mer hairpin RNA as prototypical example. This can be visualized by comparing 1D traces extracted from the conventional and HMT NOESY spectra (Fig. 3). Notice that several of the correlations are only detected in the HMT experiment, especially for iminos arising from bases that, like U7 and U8, are positioned in loop regions that undergo facile solvent exchanges. The wealth of peaks in the HMT experiment facilitates the elucidation of peaks in the amine, aromatic and sugar regions: with these, it is possible

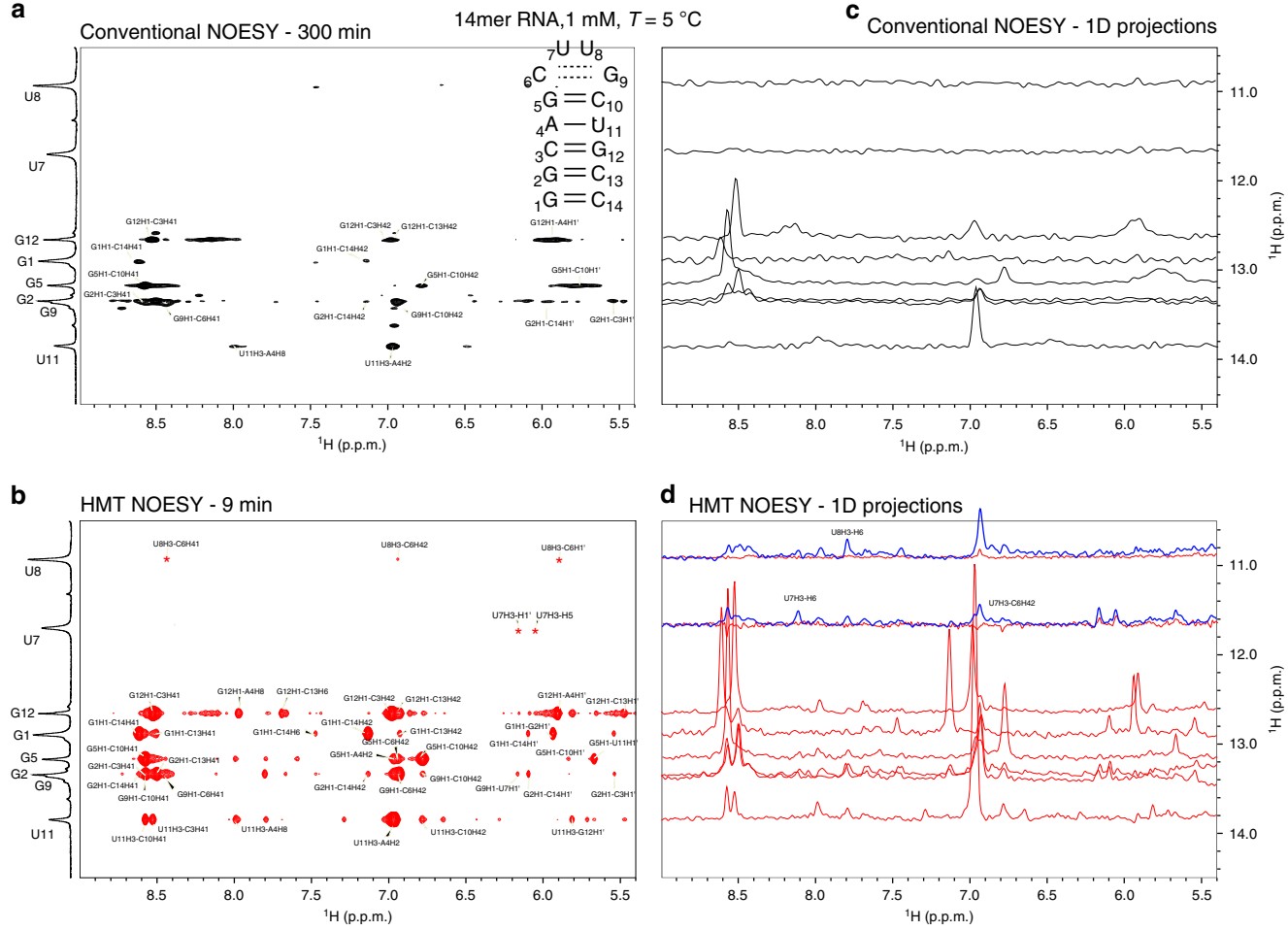

**Fig. 3 HMT applied to enhance imino-derived correlations in RNAs. a** Conventional (100 ms mixing) vs **b** HMT NOESY spectra (17 loops with a 20 Hz inversion pulse followed by 40 ms mixing each) recorded on the 14mer on top. Both 2D spectra show the imino resonances placed along the vertical direction; for maximizing its efficiency, however, the conventional spectrum was acquired with the imino resonances along the direct domain, and was flipped in this graph to match the HMT data (otherwise, if encoding the iminos in $t_1$, the conventional experiment showed no peaks altogether). As this was an [15]N-labeled sample, looped inversions/mixing periods were preferred over continuous saturations: this avoided a constant [15]N decoupling during the saturation period, which would have led to sample heating. Assignments are as labeled in the spectra; cross peaks labeled with asterisk are not visible at the given contour level. To better appreciate the enhancements, **c** and **d** show various 1D slices extracted from **a** and **b**, respectively. Shown in blue for the HMT NOESY are traces addressing U7 and U8 with a stronger (40 Hz continuous) saturation HMT scheme, revealing new correlations despite these sites' >30 s$^{-1}$ solvent exchange rates. All spectra were acquired at 1 GHz using a Bruker Avance Neo spectrometer equipped with a TCI cryoprobe.

to obtain many more spectral assignments than what is possible with conventional experiments. Also illustrated in this Figure is HMT's ability to fine-tune the extent of the saturation to the rate of solvent exchange for various labile [1]Hs—a potentially important feature for heterogeneous structures, as the tailored excitation helps overcome the fast exchange arising in flexible regions without compromising the resolving power needed to address other, more rigid regions in the RNA. Even for more slowly exchanging iminos, like U11H3, enhancements are significant, both for correlations with aromatic protons that like A4H2 can be clearly detected in the conventional experiment, as well as with labile amino protons that like C10/C3H41, barely yield cross-peaks in the conventional NOESY. A summary of the resulting enhancements for representative peaks is presented in Table 1. Comparisons of conventional and HMT NOESY experiments for this 14mer were also performed at 600 MHz; although slices extracted from these data for various imino sites also lead to substantial enhancements, the relative enhancements provided by HMT over conventional experiments were larger at 1 GHz than at 600 MHz (Table 1). This reflects one of the

aforementioned field-derived advantages: at 1 GHz, the inversion/saturation pulses addressing the labile [1]Hs can be stronger, leading to a more efficient MT and to ~25–100% larger enhancements when assessed against their conventionally collected counterparts.

Another advantage of operating at ultrahigh fields is that most amide backbone signals in small and medium-sized proteins become well resolved in 1D experiments, enabling HMT experiments on these N-bound [1]H resonances as well. Figure 4 illustrates the benefits resulting from this with comparisons between conventional and HMT NOESY acquisitions recorded on LA5, a 40-residue protein[45], and on ubiquitin, a 76-residue protein. As in the case of the RNA, these HMT experiments utilized a polychromatic looped encoding instead of single long saturation pulses, as both of these proteins were [15]N labeled and looping facilitated heteronuclear decoupling during the encoding (sequence in Fig. 1a). Extracted 1D projections show a 1:1 match between the HMT and conventional spectra, with sensitivity enhancements of ≈2x provided by the faster former scheme. These enhancements are smaller than those observed for

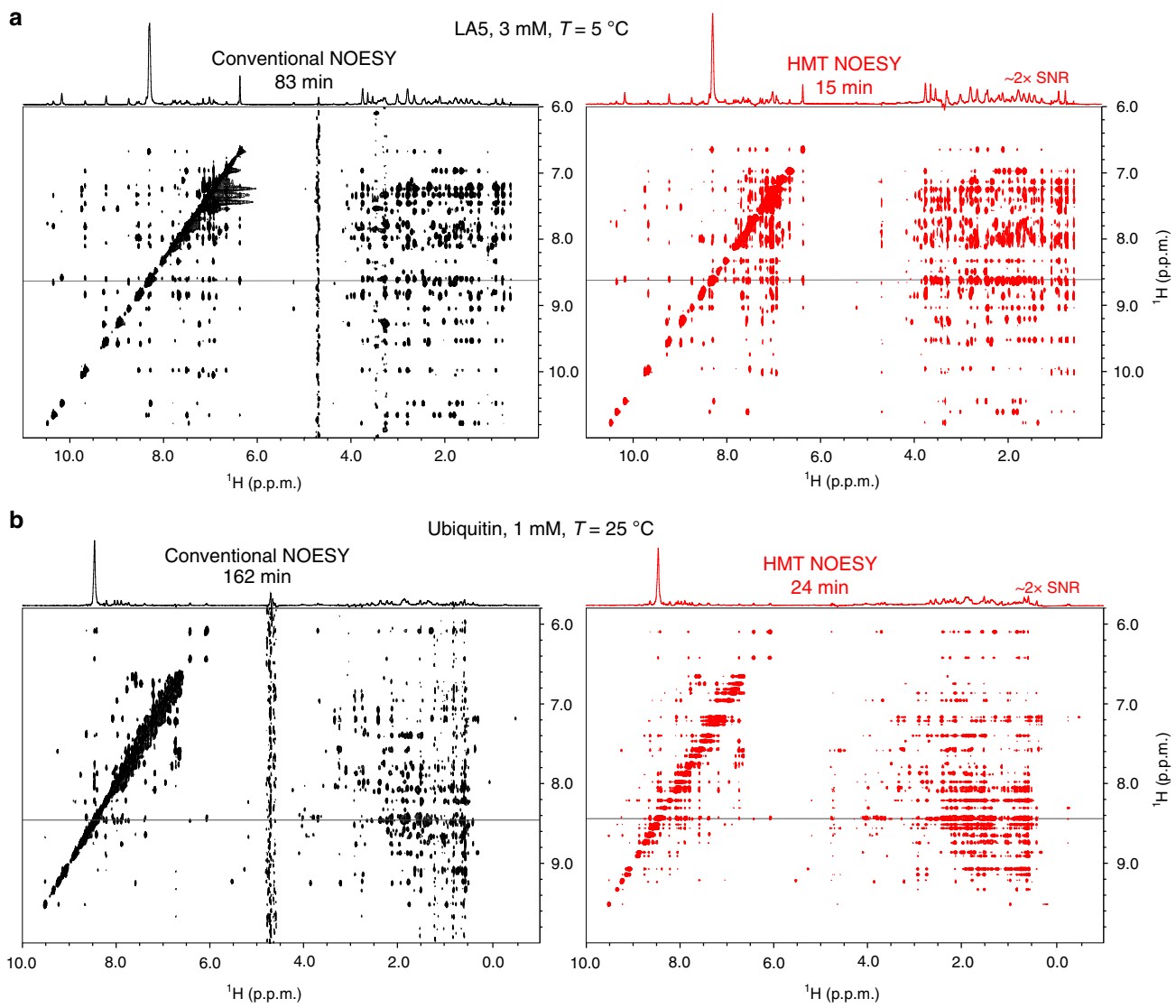

**Fig. 4 HMT applied to enhance correlations in folded proteins.** Conventional vs HMT NOESY spectra recorded for LA5 **a** and ubiquitin **b** samples. Notice how the 23.5 T field is sufficient to resolve almost entirely the amide/amino resonances in these structured peptides (regions between 6.6–9.5 ppm) and enable fast, highly sensitive NOESY experiments by Hadamard MT. Conventional experiments were acquired with 300 ms and 250 ms mixing, respectively (for maximum NOESY cross-peaks); HMT employed 6 × 150 ms and 6 × 140 ms looped encoding. Spectra were acquired at 1 GHz using a Bruker Avance Neo console equipped with a TCI cryoprobe.

saccharides and nucleic acids owing to the slower exchange rates that amide protons exhibit in the structured environments of these proteins; still when combined with fivefold shorter acquisitions, it is clear that HMT also brings substantial gains in SNR/unit_time to proteins (Table 1). Even larger enhancements could result in disordered proteins; in such cases, however, limited spectral dispersion would require the incorporation of a third, heteronuclear-encoding dimension in order to resolve the amide peaks. Such experiments will be considered in a separate study.

The strong correlations that HMT can deliver also open up the possibility of better exploiting amino groups, whose $^1$Hs are sometimes underutilized in NOESY and TOCSY experiments both in proteins and nucleic acids. Figure 5 shows how such 2D correlations can be put to good use, with HMT NOESY examples targeting amino groups in both ubiquitin and in the 14mer RNA sample. Strong correlations are observed in both spectra among the amino protons themselves, arising from combined chemical exchange and Overhauser effects among

these moieties. Similar correlations are detected, much more weakly, in the conventional experiments depicted in Supplementary Fig. 7c for ubiquitin and in Supplementary Fig. 8b for the RNA. Moreover, HMT experiments reveal many additional long-range NOE-driven cross-peaks with the aliphatic protons for the protein case, and with the imino protons of nucleotides that are both in the same and in neighboring base pairs throughout the nucleic acid chain. When compared ith the amide protons, it is clear that the faster chemical exchange of these amine sites endows their HMT data with sizable gains when considering SNR/unit_time.

**HMT: exploiting hydroxyl correlations in proteins and nucleic acids.** In addition to nitrogen-bound labile protons, hydroxyl $^1$Hs in sidechains and in sugars are notoriously challenging targets to work with in protein and nucleic acid NMR, respectively[46–48]. In both cases, these -OH peaks are often buried under other, sharper, and more intense amine and amide

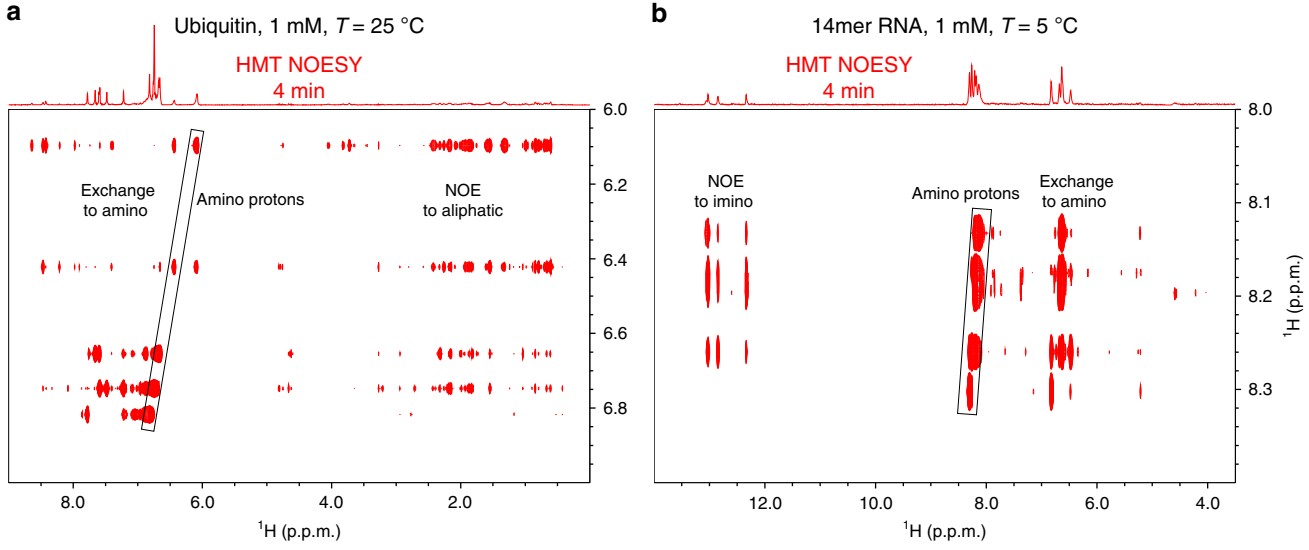

**Fig. 5 Targeting amino protons with HMT NOESY. a** Ubiquitin sample. **b** 14mer RNA in Fig. 3. The spectrum in **a** was acquired with 12 loops and 80 ms per loop, whereas in **b** a 600 ms long saturation pulse was used instead. The strongest correlations correspond to exchange and NOE cross-peaks to aliphatic and amino protons respectively; notice in the protein and the RNA, however, interesting amide→imino cross-peaks emerge as well. Spectra were acquired at 1 GHz using a Bruker Avance Neo console equipped with a TCI cryoprobe.

resonances. On the other hand, hydroxyl $^1$Hs usually undergo faster chemical exchange with water than the latter; as illustrated in Figs. 1 and 2, this qualifies them for potentially large cross-peak enhancements when targeted by the HMT scheme. Figure 6a shows a version of the HMT pulse sequence that could allow such usually "hidden" hydroxyl $^1$Hs be encoded, by incorporating additional $^{15}$N/$^{13}$C-based filters[49,50] aimed at suppressing the intense signals from $^1$Hs bound to $^{15}$N and $^{13}$C that would otherwise complicate the OH's observation. To investigate if the OH $^1$Hs could be targeted in such experiments, a series of 1D variable-temperature (5–25 °C) $^{15}$N/$^{13}$C-suppressed spectra were acquired; $^1$H resonances that survived the N-H and C-H suppression and became sharper at lower temperatures owing to slower chemical exchange with water, and were confirmed as likely candidates to arise from the labile hydroxyl $^1$Hs (Supplementary Fig. 7b). Figure 6b, c exemplify the cross-correlations arising from this experiment on a doubly $^{13}$C/$^{15}$N-labeled ubiquitin and on a 14mer RNA sample, respectively. Highlighted in these spectra are the hydroxyl $^1$Hs addressed in the HMT NOESY; particularly interesting are OH-OH inter-residue correlations detected in ubiquitin, and the long-range correlations between the OHs of sugar and the base $^1$Hs resonating between 7 and 8 ppm in the 14mer RNA that are mostly missing in the conventional spectrum (Supplementary Fig. 8c). The latter include interesting correlations between the 2′-OH groups and aromatic nucleotide sites in the 5′-direction[46,51]. Whereas in the conventional NOESY only a single such interaction is observed (between the 2′-OH group of C13 and the aromatic proton of C14), in the HMT NOESY a total of 12 such correlations were detected. These give hitherto unavailable inter-residue correlations and distance information –including interactions between the 2′-OH of C10 and the H6 of U11, 2′OH of C3 and H8 of A4, and 2′-OH of G5 and H6 of C6. Additional cross-peaks are observed, but their unambiguous assignment is still challenging in this 2D NOESY spectrum due to resonance overlap persisting even at 1 GHz. The incorporation of the HMT NOESY segment into a 3D experiment incorporating further information regarding the ribose $^1$Hs, could potentially lift such ambiguities and contribute substantially to 3D structure determinations.

## Discussion

Hadamard MT was introduced here as an extension of projective-measurement experiments where, instead of a looped $t_1$ time-domain encoding[15], selective irradiations are used to impart significant SNR/unit_time gains in homonuclear correlation experiments involving fast-exchanging protons. To achieve these gains, HMT exploits the flow of water polarization to reset the states of the targeted sites, which are then perturbed away from equilibrium via selective saturation or inversions. Polarization transfer processes spreading throughout the molecules via dipole–dipole relaxation or $J$-couplings are thus magnified, in a manner that resembles CEST-derived enhancements[52,53]. Thus, although chemical exchange with the solvent deteriorates conventional homonuclear transfer experiments, the abundant, slowly relaxing water resonance improves these processes by several-fold when switching to this new encoding scheme. Relaxation properties of the non-labile $^1$Hs on the receiving end of these transfers will limit the extent of these gains, as the MT process will only be effective over their "memory times" $T_1$. The efficiency of these experiments improves by operating at ultrahigh fields, where the labile $^1$Hs are better resolved, and the extent of the saturation-derived transfer can be increased. Possible drawbacks of relying on such high-field MT processes concern potential sample heating effects, but these were not found to be a problem even when operating at 1 GHz. Specificity in these MT experiments could be compromised by spin-diffusion among the non-labile $^1$Hs in the system; we investigated a similar possibility for the case of L-PROSY, where it was found that a 1:1 correlation with conventional NOESY cross-peak intensities was preserved even in the presence of a spin-diffusion sink pool[15]. This correspondence will persist over a wide range of solvent exchange rates and correlation times; it remains to be investigated whether the continuous-irradiation or the repeated inversion versions of HMT, might perform differently in terms of potential spin-diffusion effects. It is also interesting to note that while HMT and L-PROSY operate based on related principles, the per-scan enhancements in HMT will generally be equal to or higher than in L-PROSY. This is a consequence of: (i) the additional degree of freedom that tailoring the intensity of the saturating/inverting RF field provides to the MT experiments, and (ii) the higher per-scan

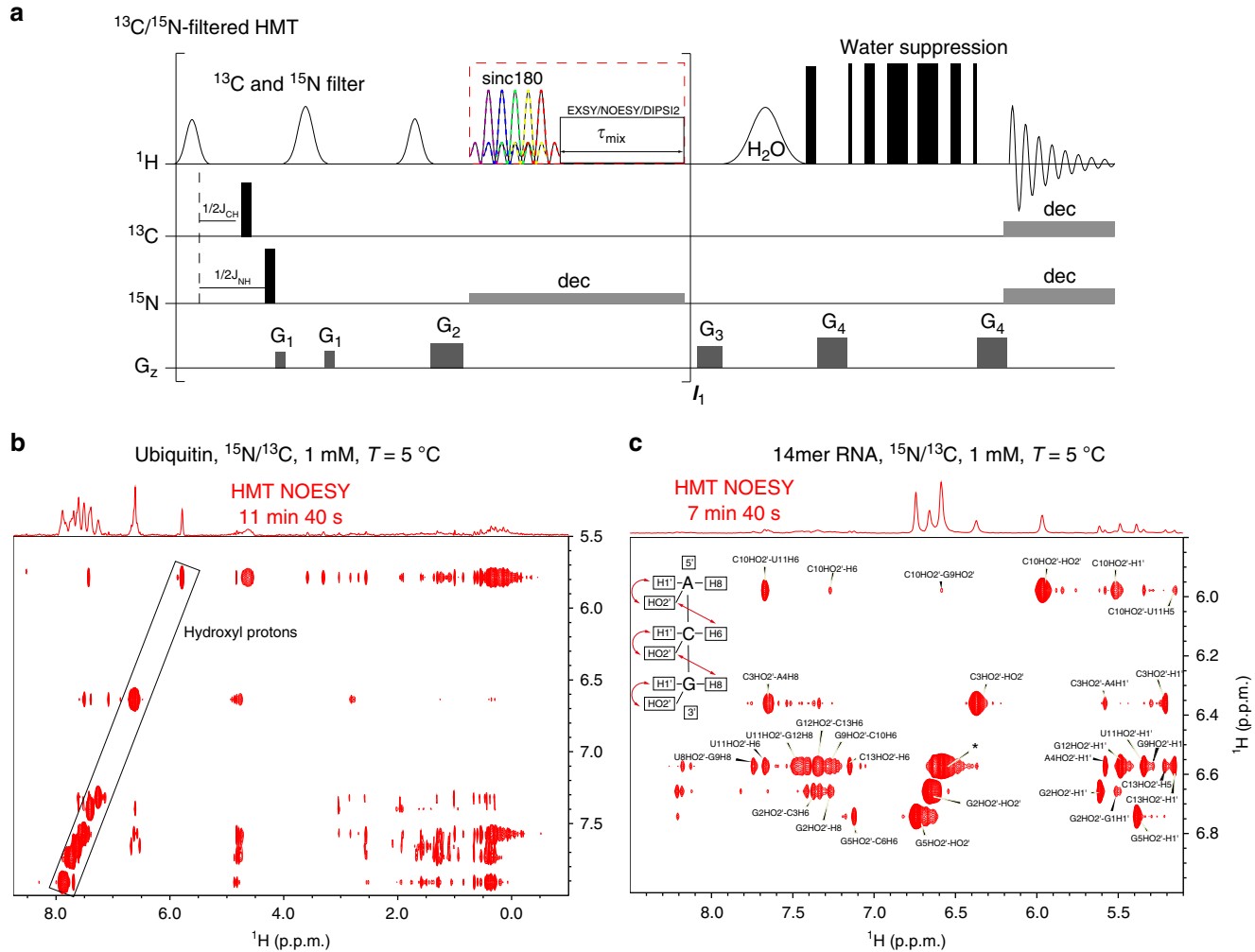

**Fig. 6 HMT incorporating a filter to suppress ¹Hs bound to ¹³C and/or ¹⁵N in labeled compounds, in order to selectively observe "hidden" hydroxyl hydrogens. a** Pulse sequence incorporating a filter is applied at the beginning of every loop in order to prevent the suppressed ¹³C- and/or ¹⁵N-bound ¹Hs to start recovering during mixing periods. Consequently, short mixing times have to be used per loop; this is not a hindrance in this case thanks to the OH's fast chemical exchange with water. In the illustrated filter, a selective ¹Hs spin-echo is applied only to the HO-bearing regions, so as not to disturb other ¹Hs that maybe receiving the magnetization from these targeted sites. Examples are illustrated with **b** ubiquitin acquired using 10 loops and 50 ms per loop, and **c** the 14mer RNA using 15 loops and 40 ms mixing time per loop. The structure in the inset illustrates the latter's expected NOE correlations involving hydroxyl ¹Hs, and a nearly complete assignment of these resonances revealing the many long-range correlations provided by the HMT spectrum. Resonances labeled with an asterisk does not have unambiguous assignment, as they involve 4 or 5 overlapping hydroxyl ¹Hs. Spectra were acquired at 1 GHz using a Bruker Avance Neo console equipped with a TCI cryoprobe.

efficiency enjoyed by frequency-based Hadamard methods over time-domain encoded counterparts, where signals decay owing to $T_2$ and exchange-driven losses over the course of the $t_1$ evolutions. On top of this there is Hadamard's "compressed sensing" time savings, whereby in the absence of signal averaging needs, the overall duration of the experiment should not require more scans than the actual number of peaks; given sufficient SNR, suitably regularized non-uniform-sampling Fourier reconstruction schemes can also enjoy from such benefits.

Although this study focused on homonuclear transfers originating from labile ¹Hs being replenished by the solvent, the HMT concept could be exploited in additional NMR settings where fast-relaxing sites can be individually addressed; in such instances, cross-peaks to slower-relaxing sites could be enhanced without a need for exchanges with the solvent. These cases may include methyl groups in otherwise deuterated proteins, small molecules interacting dynamically with bigger ones (e.g., drug–protein binding processes)[54–56], as well as fast-relaxing sites in

paramagnetic biomolecules. It is also worth exploring extension of the method to incorporate other mixing schemes, particularly those that could serve to distinguish spin-diffusion and relaxation-driven transfers from chemical exchange effects[57–59]. The high efficiency and short-acquisition of HMT experiments can be further utilized for fast multidimensional reaction monitoring, especially involving small reagents/products in the fast tumbling regime that are traditionally hard to correlate with NOESY. Last but not least, the 2D Hadamard concepts introduced here could also be included as part of correlations with heteronuclei in 3D spectral acquisitions[27,60]. Heteronuclear analogs of the homonuclear polarization transfer processes discussed here can also be envisaged. Examples of these developments will be discussed in upcoming studies.

## Methods

**Sample preparation**. Myo-inositol was purchased from Sigma Aldrich (Israel) and prepared as 5 mM solution at pH 6.0. 5 mM sucrose was prepared using household

sugar at pH 6.5. Natural abundance α2–8 (SiA)$_4$ was purchased from EY Laboratories Inc (San Mateo, CA); 25 mg of this tetramer were dissolved in 400 µL of 20 mM phosphate buffer at pH 6.5 containing 0.05% NaN$_3$, yielding a ~50 mM final solution at pH 7.35. The $^{13}$C/$^{15}$N-labeled 14mer gCUUGc tetraloop (5′-pppGGCAGCUUG-CUGCC-3′) was prepared from a linearized plasmid DNA by a run-off in vitro transcription using the T7 RNA polymerase[61]. In addition, the plasmid DNA contained a self-cleaving HDV ribozyme to ensure 3′ homogeneity. Labeled rNTPs were purchased from Silantes (Munich, Germany). The RNA was folded in NMR buffer (10 mM phosphate buffer + 1 mM EDTA pH: 6.4) in 90% H$_2$O and 10% D$_2$O by denaturing it for 5 min at 95 °C and subsequently slowly cooling down to room temperature. The final concentration of the RNA was 1 mM. Ubiquitin was purchased from Asla Biotech and was dissolved in PBS (Dulbecco's Phosphate Buffer Saline at physiological pH, purchased from Biological Industries) at a concentration of 1 mM; LA5, the ligand binding domain 5 of the low-density lipoprotein receptor LDLR, was prepared as described by Szekely et al.[45] at pH 7.4 and concentration 3 mM in 10 mM Tris buffer with 1 mM CaCl$_2$. All protein samples were prepared in H$_2$O/D$_2$O (90%:10%) solutions containing NaN$_3$. Although the use of $^{13}$C/$^{15}$N spin labels in these protein and RNA samples was not needed within the context of this study, they were used as a result of their immediate availability. It is worth noting that, because of this labeling, efficient composite pulse or adiabatic decoupling could be implemented throughout the acquisition of all 2D NOESY/TOCSY data (rather than just during $t_2$ and at $t_1$/2).

**NMR experiments**. NMR experiments were conducted using either a 14.1 T Bruker magnet equipped with an Avance III console and TCI Prodigy probe; or a 1 GHz, 23.5 T Bruker Avance Neo equipped with a TCI cryoprobe. Hadamard experiments were carried out using 8x8 through 64 × 64 Hadamard encoding matrices, depending on the number of peaks in the spectrum. In total, 10–40 Hz nutation fields were used for saturation, whereas 20–25 Hz bandwidth *sinc1* pulses were used in looped inversion method. The number of loops and/or duration of the saturation, were determined according to T$_1$ values of the receiving protons. Optimal values for NOESY and TOCSY mixing times were used in conventional experiments. Conventional TOCSY were acquired using dipsi2gpph19/dipsi2esgpph Bruker sequences employing DIPSI2 isotropic mixing[13], whereas for NOESY experiments noesyfpgpph19/noesyesgpph was used employing Watergate 3919[62] with a flip-back pulse[63], or excitation sculpting for water suppression. For most of the compounds, Watergate 3919 with water flip-back pulses showed excellent performance in keeping the water unperturbed[31,64–66], and was preferred over jump-return techniques owing to the larger bandwidth of their binomial excitation. In all comparisons between HMT and conventional NOESY or TOCSY acquisitions, the experiments differed only in their encoding and mixing principles; water-suppression techniques, recovery delays d$_1$ and receiver gains were otherwise set identical, in order to ensure faithful comparisons. Mixing times and Watergate delays for the binomial water suppression were optimized according to the magnetic field. All spectra were processed in Bruker® TopSpin® 4.0.6 and 4.0.9. All spectra were apodized with QSINE or SINE window functions and while conventional spectra were zero-filled once, all Hadamard spectra were zero-filled to 256 × 1024 points. Spectra were analyzed using NMRFAM-SPARKY[67].

**Reporting summary**. Further information on research design is available in the Nature Research Reporting Summary linked to this article.

## Data availability

The data sets generated and analysed during the current study are available from the corresponding author on reasonable request.

## Code availability

HMT sequences were deposited in an RNA-specialized Bruker NMR User Library, and its details/uses further explained in the Bruker-sponsored website https://www.bruker.com/service/information-communication/nmr-pulse-program-lib/bruker-user-library/liquids/avance-neo.html. Parameters there are given for the imino-based experiments; adjustments to target other kinds of $^1$Hs may require adjusting offsets and powers.

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

## Acknowledgements

We are grateful to professor D. Fass (Weizmann Institute) for the LA5 sample and to Dr. Tali Scherf (Weizmann Institute) for the assistance in the GHz experiments. This work was supported by the Kimmel Institute for Magnetic Resonance (Weizmann Institute), the EU Horizon 2020 program (FET-OPEN Grant 828946, PATHOS), Israel Science Foundation Grant 965/18, and the Perlman Family Foundation. H.S. was supported by DFG-funded collaborative research center 902. Work at BMRZ is supported by the state of Hesse. Joint support to L.F., H.S. was given by the German-Israel Foundation (grant G-1501-302). We wish to thank Boris Fürtig, Robbin Schnieders, and Christian Richter for stimulating discussions.

## Author contributions

M.N., E.K., and L.F. conceived the project. M.N., E.K., and L.F. implemented the method. M.N. carried out the NMR measurements and the spin simulations. M.N. and E.K. processed the data. M.N., A.O., H.S., and L.F. conceived the nucleic acids applications and evaluated their NMR data. M.N., M.D.B., D.F., and L.F. conceived the oligo-saccharide applications and evaluated their NMR data. M.N. and L.F. wrote the paper. All authors contributed to the discussions leading to the final manuscript.

## Competing interests

Dr. Eriks Kupce is an employee of the Bruker Corporation. The remaining authors declare no competing interests.
