## [Peer Review File · Nature Communications]

REVIEWER COMMENTS

Reviewer #1 (Remarks to the Author):

This manuscript reports a significant advance in NMR data acquisition methods commonly used in structural characterization of proteins, nucleic acids and carbohydrates. It builds on proton exchange phenomena that can tap a pool of magnetization stored in solvent water to dramatically improve sensitivity. This store has been tapped before in experiments like CEST and in the author's own previous publication on looped methods (L-PROSY). However, here Hadamard acquisition and processing have been added to achieve enhancements that reach 6-8 fold. There is an extensive set of illustrations on sugars, nucleic acids and proteins. Although the major point could be made with fewer illustrations, different examples may gain the attention of a broader audience.

There are a few points that might be clarified or at least given more emphasis. One is that the method works for a subset of sites normally observed in target molecules, namely OH and NH groups where protons exchange rapidly with protons in water. This limits applicability, but at the same time adds data normally missed. The added data often involve groups involved in structurally important hydrogen bonds. A good illustration of this importance may be more useful than the extensive group of illustrations.

Another limitation is lies in the selective pulses used for Hadamard excitation. This limits resolution and may introduce some ambiguity in NOE and other contacts. Fig 1b, for example shows some bleed over between excitation at 5.85 and 5.82, and there are some noticeable horizontal stripes in figure 4b. The authors might comment on this limitation. Also, specific data are not given on the Hadamard excitations until a general mention of H8 - H64 is made in the methods section. Specific statements about what is used could easily be added to the figure legends.

Reviewer #2 (Remarks to the Author):

The paper describes a general applicable NMR method that improves the sensitivity of observing protons in (bio-)molecules that rapidly exchange with water. The obtained sensitivity improvement is impressive. The basic idea and certainly its implementation - combining 2D Hadamard sampling minimising solvent excitation and saturation - is elegant, but these concepts are not really new.

Despite demonstration to several examples, the manuscript does not lead to new insight in structure, function or dynamics of the (bio-)molecules studied and therefore lacks a convincing demonstration of the potentialities of this NMR method to mark it as a breakthrough that interests a broad readership. Novel observation / demonstration of a molecule or process that could not be studied before is lacking. Have we learned something new of myo-inositol or RNA that we did not know before? Without this the current manuscript appears only an incremental advance over L-PROSY in 2018, and may only be of interest for the biomolecular NMR community.

Technical remarks:

- The concept is based on combining Hadamard sampling with avoiding solvent saturation. In fact the improved sensitivity comes largely from the last. The first is well cited (with articles by one of the co-authors), but I miss citations of original literature that addressed avoiding water saturation to enhance NMR sensitivity (Redfield, Gueron, etc; when summarized through reviews, this could be indicated).
- p.9 use of isotope filters to suppress non-exchangeable NH and CH protons to allow detection of hydroxyls is not new. This application misses original citations.
- Fig. 1b. A conventional NOESY is not the proper reference for demonstrating improved observation of NOEs of labile protons. It is well-known that conventional NOESY's are insensitive and need to be replaced by NOESY that use semi-selective pulse that do not excite water.
- The pulse sequence "noesygpph19": Watergate with use of gradients is very effective in

suppressing solvent signals, but at the expense of the water signal intensity. The water signal will be effectively saturated, thus this severely hampers the sensitivity of this NOESY, and appears no fair comparison to the Hadamard approach. JR or other schemes, that avoid exciting solvent, should be a better comparison.

- p.4 I miss mention of cross-peaks to water. Were they absent? In that case the hydroxyls are not so labile after all.

Minor:

- (Methods, lacking detail). How were the conventional NOESY's recorded? How was the solvent signal suppressed (when presat which RF field strengths and which duration, etc). Were spectra recorded with the same receiver gain settings, etc? What are the actual sensitivity gains in the NOESY's (p.7)?

- Transition p.3/4. Constant repolarization: less unique than it looks. Gueron's JR is an old idea, even many methods existed before that. Observation of hydroxyl protons has also been achieved well before.

- p.5 water exchange rate up to 100 s⁻¹. What is NOE mixing time: when 100 ms, all exchangeable proton intensity will have been transferred to water, without possibilities for NOEs with any other protons.

- Fig 2b. Does the differential linewidth correspond to water exchange rates?

Very minor:

p.5 wide rate of..? wide range of..

p.6 ..than what possible...? than was possible..

p.12 ...noesyfgpph19 ? or: noesygpph19

Reviewer #3 (Remarks to the Author):

As it follows from the title, the manuscript presented by Mihajlo Novakovic and co-authors introduces a new method for sensitivity enhancement of NMR peaks originating from labile protons in proteins, polysaccharides and nucleic acids. The result is a follow up of an earlier work of the same group (Novakovic, et al., J. Magn. Reson. 294, 169–180, 2018), which introduced the Looped-PROjected Spectroscopy (L-PROSY). The main technical improvement of the current work is combination of the L-PROSY approach with a measurement time saving technique Hadamard spectroscopy. This allows to drastically reduce time of the experiment without sacrificing its quality and information content. Other advantages of using the Hadamard over the original chemical shift encoding scheme in the L-PROSY includes additional sensitivity enhancement and elimination of some spectral artefacts. The most exciting part of the work that clearly sets a new landmark in the NMR and the broader (bio)chemistry field is an array of impressive demonstration of the new method on a wide range of biomolecular systems: proteins, polysaccharides and nucleic acids. The authors show that so far practically inaccessible labile (i.e. rapidly exchanging with water) protons can now be used as valuable sensitive probes in two major types of NMR experiments. Thus, the paper should be of broad interest and deserves publication in NM after addressing several of minor issues described below.

1. Among methods for saving measurement time in multidimensional NMR experiments, the Hadamard approach is a credible although not the most commonly used. It would be good to see a short discussion on why the Hadamard is preferred in the work in comparison, for example, to the NUS or single-scan NMR? In particular, Hadamard has a notable practical disadvantage because of the requirement that the target peaks must be well separated and their positions are known in advance.

2. It is not clear how were the presented 2D spectra reconstructed in the Hadamard dimension? In particular, what is the meaning of the line shape in this dimension. In other words, are the 2D spectra real or just a 2D-like presentation of the 1D traces at the selected (known) target frequencies?

3. Several examples in the paper, where the rapidly exchanging with water amide or amino protons were targeted, were performed on the $^{15}\text{N}/^{13}\text{C}$ labelled sample. This prevented use of the simpler and probably more robust saturation version of the NOESY experiment. It should be explained why did the authors use such samples, if the new experiment made no use of the labelling.
4. The authors should clarify individual contributions of the L-PROSY and Hadamard parts of the new the HMT method. In the presented examples both the L-PROSY and HMT spectra show high peak intensity enhancement relative to the traditional experiments. However, although the HMT methods require much less measurement time, it is not clear if Hadamard is merely a time saving tool or does it bring, and if so how much, an additional gain in the sensitivity per unit time relative to the core L-PROSY approach.
5. Authors should share the pulse sequences introduced in the work (Fig. 1) and give more details about the experiment setup.

RE: Revised version of Nature Communications manuscript

As can be appreciated from the “responses” below, we have subjected our manuscript to a revision that addressed all major and minor critiques and suggestions raised by the manuscript’s reviewers. The paragraphs below describe in a point-by-point fashion the actions taken in response to the requests and criticisms raised by the Referees. For ease of tracking we describe first in black font the individual critiques that were raised; these are followed by a blue font paragraph summarizing the action taken in response to each critique. For completion we are also submitting “highlighted” pdf versions of the manuscript and the supporting information remarking all the changes we made; however, as these are numerous and sometimes make it difficult to concentrate on the text itself, we are also uploading “clean” versions of the revised paper’s main text & figures and of its SI for the sake of reviewing. Furthermore, we are also uploading clean Word versions of these files.

Responses to the Referees’ comments

Reviewer #1:

This manuscript reports a significant advance in NMR data acquisition methods commonly used in structural characterization of proteins, nucleic acids and carbohydrates. It builds on proton exchange phenomena that can tap a pool of magnetization stored in solvent water to dramatically improve sensitivity. This store has been tapped before in experiments like CEST and in the author’s own previous publication on looped methods (L-PROSY). However, here Hadamard acquisition and processing have been added to achieve enhancements that reach 6-8 fold. There is an extensive set of illustrations on sugars, nucleic acids and proteins. Although the major point could be made with fewer illustrations, different examples may gain the attention of a broader audience.

We thank the expert for this enthusiastic overall assessment. Indeed, as s/he notices, it is the combined action of an RF-regulated water→labile→non-labile transfer and a Hadamard strategy encoding these repolarization/cross-relaxation processes in minimal number of steps, that end up enhancing sensitivity by severalfold –allowing in turn to accelerate the NMR experiments accordingly. We are also happy to see that the expert agrees with our choice of demonstrating these enhancements on a wide array of targets, to better engage the potential audience and users.

There are a few points that might be clarified or at least given more emphasis. One is that the method works for a subset of sites normally observed in target molecules, namely OH and NH groups where protons exchange rapidly with protons in water.. This limits applicability, but at the same time adds data normally missed. The added data often involve groups involved in structurally important hydrogen bonds. A good illustration of this importance may be more useful than the extensive group of illustrations.

This is a good point, which we have tried to address by better describing the limits of HMT’s applicability. For instance the inclusion of the imino-imino 2D regions in Fig. 3 may have given the impression that labile↔labile correlations between these sites could be extracted by HMT; this is not the case. These and similar points were further sharpened in the revised manuscript (also in connection to the other referees’ critiques, *vide infra*). We have also included a new Table, summarizing the enhancements/unit_time achieved for the different systems.

Another limitation lies in the selective pulses used for Hadamard excitation. This limits resolution and may introduce some ambiguity in NOE and other contacts. Fig 1b, for example shows some bleed over between excitation at 5.85 and 5.82, and there are some noticeable horizontal stripes in figure 4b. The authors might comment on this limitation.. Also, specific data are not given on the Hadamard excitations until a general mention of H8 - H64 is made in the methods section. Specific statements about what is used could easily be added to the figure legends.

This is a good point, also mentioned by Reviewer #3 in his/her opening statements. We had tried to stress this – for instance by showing the importance of relying on high-field NMR for doing these experiments (e.g., Fig. 2). We have revised the text and enriched the figure captions, in the hope that this limitation is better noticed. We

have also clarified further the nature of the “bleed through”, by deriving something akin to the indirect-domain point spread function for these experiments. As for the bleed through that as the expert remarks may arise in case of overlapping resonances: it is indeed there but it is less critical than what it may appear –primarily due to the labile nature of the 1Hs whose 2D correlations these new experiments target. This lability usually leads to exchange-broadened 1D linewidths, whose spectral resolution is not much further compromised by the saturation/inversion processes used in their HMT encoding. In fact, one of the advantages of our new experiment is its ability to tailor the saturating field / inversion bandwidth to each spectral peak individually, enabling an optimal compromise between resolution, and the extraction of the information being sought. Further details are given below.

Reviewer #2

The paper describes a general applicable NMR method that improves the sensitivity of observing protons in (bio-)molecules that rapidly exchange with water. The obtained sensitivity improvement is impressive. The basic idea and certainly its implementation - combining 2D Hadamard sampling minimising solvent excitation and saturation - is elegant, but these concepts are not really new.

We are afraid that we failed to convey to this expert the essence of the experiment. While combining elements that as noticed are not new –Hadamard encoding, selective saturation pulses, mixing times– HMT does lead to a way of obtaining NOE/J-correlations that is essentially different from that of NOESY or TOCSY. Perhaps the best way of intuitively conveying this difference is by making an analogous comparison between CEST and 2D EXSY: while both of these experiments provide insight and rates about interconverting species, the former will *under the right conditions*, do so with a multi-fold sensitivity enhancement. Indeed, there is little to be gained from CEST if the “starting” (irradiated) spin in the two-site exchange process has the longer T1, and/or if the exchanging site populations are nearly equal. Likewise, there would be little to gain from measuring cross-relaxation connectivities using saturation-based principles instead of a conventional NOESY mixing, *unless (i) the originating 1Hs are labile and hence rapidly repolarizing, and (ii) if the originating 1Hs wouldn't have a low intensity due to exchanges with the solvent, while the receiving ones were not*. These two features apply to the cases analyzed in our contribution, endowing them with CEST-like advantages when probing their cross-correlations with an MT-based scheme, over a NOESY/TOCSY mixing scheme. Couple to this the fact that, unlike what happens in bioCEST experiments where all frequencies need to be probed step-by-step, the positions of the “starting” labile sites are *a priori* known and hence can be Hadamard-encoded, plus the fact that each of this site’s saturation can be individually tailored, and our new scheme ends up enhancing the sensitivity/unit_time of labile-derived cross-peaks by several hundred-folds.

In an effort to make these NOESY/TOCSY vs HMT differences more clear, a main addition we have done to the revised manuscript includes an analytical derivation of the cross-peak signal intensities for the two schemes. The end results of these derivations make the dependence of HMT’s gains on the various sites T1, T2, solvent exchange rates and saturation power clear –and clearly different of the gains associated to a NOESY/TOCSY experiment, regardless of whether the latter employs solvent suppression or not.

Despite demonstration to several examples, the manuscript does not lead to new insight i, function or dynamics of the (bio-)molecules studied and therefore lacks a convincing demonstration of the potentialities of this NMR method to mark it as a breakthrough that interests a broad readership. Novel observation / demonstration of a molecule or process that could not be studied before is lacking. Have we learned something new of myo-inositol or RNA that we did not know before? Without this the current manuscript appears only an incremental advance over L-PROSY in 2018, and may only be of interest for the biomolecular NMR community.

We respectfully disagree with this assessment: “quantity has a quality of its own”, and reducing an experiment’s duration by several hundred-fold while at the same time evidencing peaks that were previously lost in noise is –as solid state DNP NMR practitioners will attest– far from an incremental advance. Even for a molecule as simple as myo-inositol, the possibility now opened to measure its OH cross-correlations has opened valuable *in vivo* opportunities that we are starting to explore. Furthermore, as pointed out by Reviewer #1, it was our goal to keep generality rather than go into solving a single, specific problem.

Still, driven by the paragraph above, we expanded onto our tetraglycan analysis and now show that the cross-

relaxation peaks that HMT reveals for it in water at 5 °C, impose considerable changes to the structure that had been previously proposed on the basis of -10 °C non-labile NOE measurements recorded at lower fields. This is further explained in revised Fig. 2 and its associated discussion and is a change that arises, among other reasons, from the structure's temperature dependence. We are observing similar behaviors in the structures of multi-bulged RNAs (data not shown), where both average structure and inter-strand dynamics are deeply affected by temperatures, and structures at (physiological) temperatures that can rarely be studied by conventional NOESY become amenable to the HMT experiments. Clearly, the appearance of previously invisible NOE cross-peaks is liable to change the structural conclusions that can be obtained from solution NMR.

Some Technical remarks:

- The concept is based on combining Hadamard sampling with avoiding solvent saturation. In fact the improved sensitivity comes largely from the last. The first is well cited (with articles by one of the co-authors), but I miss citations of original literature that addressed avoiding water saturation to enhance NMR sensitivity (Redfield, Gueron, etc; when summarized through reviews, this could be indicated).

As mentioned, we seemed to have failed to convey the fact that it is not avoiding saturation *per se* what enhances this experiment's sensitivity, but rather the fact that exchanges with the solvent enable the RF irradiation of labile sites to act as a "valve" in a solvent→labile→non-labile 1H polarization conveyor. As the frequency of this irradiation matches that of the *a priori* known labile 1H positions and site-specific cross-relaxations drive the final step in this chain, the intensity changes that then affect the non-labile resonances can be translated by a Hadamard transform into NOESY/TOCSY-like "cross-peaks". Hopefully the revised text and the inclusion of an analytical treatment, better reflects this idea.

Regardless of this comment, we have added the suggested references to the revised manuscript.

- p.9 use of isotope filters to suppress non-exchangeable NH and CH protons to allow detection of hydroxyls is not new. This application misses original citations.

References were expanded as requested (30, 31, 65-68 in revised ms, plus references to SOFAST-type experiments).

- Fig. 1b. A conventional NOESY is not the proper reference for demonstrating improved observation of NOEs of labile protons. It is well-known that conventional NOESY's are insensitive and need to be replaced by NOESY that use semi-selective pulse that do not excite water.

- The pulse sequence "noesygp19": Watergate with use of gradients is very effective in suppressing solvent signals, but at the expense of the water signal intensity. The water signal will be effectively saturated, thus this severely hampers the sensitivity of this NOESY, and appears no fair comparison to the Hadamard approach. JR or other schemes, that avoid exciting solvent, should be a better comparison.

The NOESY sequence we chose for our comparisons is the most commonly used one, and hence we considered it fairest to use it. The WG3919 with flipback block that used in these NOESY acquisitions performed very similarly to the JR scheme, while showing considerably better flexibility in exciting the relatively wide spectral ranges that we had to cover at 1 GHz. In the RNA 14mer spectrum illustrated below for instance, virtually no differences can be found between the two modes; this is not altogether surprising, given that both experiments strived to preserve water along the z-axis most of the time, that a long recycle delay (2 sec) is used in both cases, and that their encoding's are identical (solely differing right before the t2 acquisitions):

Only for the (SiA)4 case water couldn't be preserved as perfectly as needed due to the proximity between the water peak and the OH resonances; this forced us to use excitation sculpting. Still, also in this case, both the HMT and NOESY experiments used the same excitation sculpting scheme; in all cases, we did our best to compare apples to apples...

- p.4 I miss mention of cross-peaks to water. Were they absent? In that case the hydroxyls are not so labile after all. Since as mentioned in the text all our acquisitions included a final water-suppression block, there was no way we could detect cross-peaks to water. Although CEST-like experiments certainly show that they exist.

Minor:

- (Methods, lacking detail). How were the conventional NOESY's recorded? How was the solvent signal suppressed (when presat which RF field strengths and which duration, etc). Were spectra recorded with the same receiver gain settings, etc? What are the actual sensitivity gains in the NOESY's (p.7)?

Again: we may have not stressed enough the importance of preserving the aqueous solvent. Certainly no presat was performed in our experiments! And although we refer to our acquisitions as using "solvent suppression", we should have highlighted better that water was actually preserved longitudinally throughout most of the acquisitions. We have edited the revised manuscript to correct for this. As for the receiver gain: given the high dynamic range architecture of the new Bruker Neo consoles used in the experiments and the quality of the water preservation, receiver gain could always be set to be nearly maximal. This is now clarified in Methods. Finally, as for the sensitivity gains – there are certainly *many* cross-peaks in the data that the manuscript shows, and we didn't quantify them all. But we have compiled a new Table (#1) summarizing Enhancement/unit_time for the various experiments and samples studied, using a sizable number of representative cross-peaks.

- Transition p.3/4. Constant repolarization: less unique than it looks. Gueron's JR is an old idea, even many methods existed before that. Observation of hydroxyl protons has also been achieved well before.

As mentioned, a number of relevant citations were added to this paragraph to account for this critique.

- p.5 water exchange rate up to 100 s⁻¹. What is NOE mixing time: when 100 ms, all exchangeable proton intensity will have been transferred to water, without possibilities for NOEs with any other protons.

The reviewer brings up one of HMT's strong points vs NOESY: when it comes to labile 1Hs, NOESY's "one size fits all" framework just may not be good enough. There will be an optimal mixing time for sites like SiA4's OH4 exchanging at ≥ 100 s⁻¹ which won't lead to strong cross-peaks for the OH8 site exchanging at ~ 10 s⁻¹ –and viceversa. In the figure in question we chose a mixing that suited the latter, as it matched values normally used in these experiments (should we have chosen a shorter mixing favouring the faster-exchanging sites we would have probably been faulted of choosing a biased choice as well □). Such constraint usually forces the acquisition of multiple, protracted 2D NOESY acquisitions for fully characterizing this kind of systems. By contrast HMT's information *plateaus* at a time defined by the receiving, non-labile 1H T1s (see for instance Supporting Figures S3, S4), and hence the longest saturation time will fit all cases. Furthermore, as mentioned above, the HMT approach opens the possibility of using different saturation / inversion fields for different labile sites, as suited according to their exchange

rates (and as tunable by apparent inspection of these sites' linewidths; see comment below). We have tried to further clarify all these matters in the revised version.

- Fig 2b. Does the differential linewidth correspond to water exchange rates?

Exactly – with some field-dependent scaling factor that may vary with field. But by visual inspection of the data one can already decide which peaks should be irradiated with 10, with 20 or with 40 Hz γB_1 fields for maximizing the ensuing cross-peak HMT information

Very minor:

p.5 wide rate of..? wide range of..

p.6 ..than what possible...? than was possible..

These points were accounted for in the revision

p.12 ...noesyfgpph19 ? or: noesygpph19

noesyfgpph19 was indeed the sequence used, as this is the version incorporating the water flip back; noesygpph19 has WG3919 but without flipback, and hence sub-optimal for our experiment.

Reviewer: 3

As it follows from the title, the manuscript presented by Mihajlo Novakovic and co-authors introduces a new method for sensitivity enhancement of NMR peaks originating from labile protons in proteins, polysaccharides and nucleic acids. The result is a follow up of an earlier work of the same group (Novakovic, et al., J. Magn. Reson. 294, 169–180, 2018), which introduced the Looped-PROjected Spectroscopy (L-PROSY). The main technical improvement of the current work is combination of the L-PROSY approach with a measurement time saving technique Hadamard spectroscopy. This allows to drastically reduce time of the experiment without sacrificing its quality and information content. Other advantages of using the Hadamard over the original chemical shift encoding scheme in the L-PROSY includes additional sensitivity enhancement and elimination of some spectral artefacts. The most exciting part of the work that clearly sets a new landmark in the NMR and the broader (bio)chemistry field is an array of impressive demonstration of the new method on a wide range of biomolecular systems: proteins, polysaccharides and nucleic acids. The authors show that so far practically inaccessible labile (i.e. rapidly exchanging with water) protons can now be used as valuable sensitive probes in two major types of NMR experiments. Thus, the paper should be of broad interest and deserves publication in NM after addressing several of minor issues described below.

These are truly encouraging comments; thanks! We've tried to address this expert's remaining requests, as follows

1. Among methods for saving measurement time in multidimensional NMR experiments, the Hadamard approach is a credible although not the most commonly used. It would be good to see a short discussion on why the Hadamard is preferred in the work in comparison, for example, to the NUS or single-scan NMR?? In particular, Hadamard has a notable practical disadvantage because of the requirement that the target peaks must be well separated and their positions are known in advance.

The disadvantage noted by the referee is indeed there –and we stress it as well in the manuscript. On the other hand, and as explained in more detail above and hopefully made clearer in the revised manuscript, lying at the core of the HMT experiment is the CEST-inspired manner by which it establishes NOESY- and TOCSY-like correlations. This option is solely available if using continuous-wave-like approaches, whose “on/off” nature makes them liable to multiplexing by relying on Hadamard but not on other (phase-modulated) encoding approaches. In this case, we believe it is clear that by enabling HMT's transfer mechanism to work, the disadvantage rightly-noted above becomes outweighed by the method's ensuing sensitivity gains. Incidentally, notice that sensitivity/unit_time comparisons like those involving the SiA4 glycan *had* been performed against 25% NUS NOESY acquisitions.

- 2. It is not clear how were the presented 2D spectra reconstructed in the Hadamard dimension? In particular, what is the meaning of the line shape in this dimension. In other words, are the 2D spectra real or just a 2D-like presentation of the 1D traces at the selected (known) target frequencies?

As happens with other compressed-sensing approaches (including NUS and all kind of regularized-based reconstructions) the spectra that we generate are indeed representations that best match the collected data. This, however, does not mean that they lack a point-spread function along the “indirect” domain: in this case it will be given by either: (i) a Lorentzian whose effective width combines the natural linewidth, T_1 , the exchange broadening, and the broadening introduced by the selective RF if using CW irradiation as now described by our analytical derivations in the revised supporting information; or (ii) by squares whose widths reflect the bandwidth used in the selective Sinc pulses (or whatever the Fourier conjugates of the pulses are) in the pulsed inversion version. These matters are now clarified in the revised manuscript

3. Several examples in the paper, where the rapidly exchanging with water amide or amino protons were targeted, were performed on the $^{15}\text{N}/^{13}\text{C}$ labelled sample. This prevented use of the simpler and probably more robust saturation version of the NOESY experiment. It should be explained why did the authors use such samples, if the new experiment made no use of the labelling.

The referee is entirely correct in that there was no need in having either the RNA or protein samples labeled for this study; their use was a matter of convenience and availability. A point clarifying this has been included in the revised manuscript. Also worth noting, however, is that an efficient composite pulse decoupling was implemented throughout the acquisition of all 2D NOESY/TOCSY data (rather than just during t_2 and at $t_1/2$) for the sake of keeping a fair comparison. And while the decoupling implemented during the Hadamard encoding was arguably less efficient than in the pulsed counterparts as it was performed at lower powers to avoid heating over to the relatively long saturation times and high fields involved, this was –again– as fair a comparison as we managed to make.

4. The authors should clarify individual contributions of the L-PROSY and Hadamard parts of the new the HMT method. In the presented examples both the L-PROSY and HMT spectra show high peak intensity enhancement relative to the traditional experiments. However, although the HMT methods require much less measurement time, it is not clear if Hadamard is merely a time saving tool or does it bring, and if so how much,, an additional gain in the sensitivity per unit time relative to the core L-PROSY approach.

HMT and L-PROSY operate on the basis of related –but not identical– principles. Still the expert is correct in that: (i) by looping its selective encoding, a properly timed L-PROSY experiment manages to overcome NOESY’s “one size fits all” mixing time problem alluded to earlier; (ii) in parallel with the HMT case, L-PROSY’s cross-peaks will also plateau to intensities depending on the non-labile ^1H T_1 s. Nevertheless, as can be appreciated from the data in the SI, HMT always provides *per scan* enhancements that are equal or higher than L-PROSY. This is the consequence of two main factors. One is the additional degree of freedom involved in the new MT experiments, as given by the intensity of the saturating RF field / repetition time of the inverting pulse. The other comes from the higher per-scan efficiency enjoyed by Hadamard over Fourier encoding, when targeting labile, fast-exchanging sites. Indeed, both of these methods should in principle improve SNR as $\sqrt{N/2}$ –where N is the number of peaks in the Hadamard case and of number of t_1 increments in the Fourier case (i.e., each experiment’s number of scans before averaging), while the common 0.5 factors reflect the fact that modulations only extend between 0 and 1 when saturating in HMT (this 0.5x loss can be avoided in HMT if its encoding is performed using inversions instead of saturations –but we shall not consider that) and that on average half the information is lost in a Ramsey-type t_1 modulation. *However*, whereas in Fourier-encoded experiments like L-PROSY the t_1 -modulated signal rapidly decays as a combined result of T_2 relaxation and exchange-driven decoherence, no such hit is taken by the Hadamard multiplexing of its MT-based experiments. Encodings in this latter instance are longitudinal and hence less affected by transverse relaxation losses, leading to an added per-scan gain. On top of all this there is of course Hadamard’s “compressed sensing” time savings, whereby in the absence of signal averaging needs the overall duration of the experiment does not require more scans than the actual number of peaks.

We have strived to clarify all these important aspects in the Discussion of the revised manuscript, as they were not fully explained in the original submission.

5. Authors should share the pulse sequences introduced in the work (Fig. 1) and give more details about the experiment setup.

In fact, while this manuscript was being reviewed, the HMT sequences were deposited in an RNA-specialized Bruker NMR User Library, and its details / uses further explained in the Bruker-sponsored website

<https://www.bruker.com/service/information-communication/nmr-pulse-program-lib/bruker-user-library/liquids/avance-neo.html>. We did this because HMT has become a prime tool in our ongoing SARS-CoV-2 RNA studies, and we thought that the Covid-19 structural biology community could benefit from it. In the meantime, we are working on depositing a sequence with more extensive comments and flags, so that also polysaccharide and protein labeled and unlabeled samples can be tackled in a user-friendly fashion. This is to become available by the end of this manuscript's reviewing process.

We would like to conclude by thanking the Referees again for their comments –which we view as most valuable and constructive. We trust that our clarifications and additions will have satisfied their concerns, and make the revised study suitable for publication in *Nature Communications*.

Thanks in advance for your attention regarding this matter.

Lucio Frydman